# Differences in Leaf Morphology and Related Gene Expression between Diploid and Tetraploid Birch (*Betula pendula*)

**DOI:** 10.3390/ijms232112966

**Published:** 2022-10-26

**Authors:** Xiaoyue Zhang, Kun Chen, Wei Wang, Guifeng Liu, Chuanping Yang, Jing Jiang

**Affiliations:** State Key Laboratory of Tree Genetics and Breeding, Northeast Forestry University, 26 Hexing Road, Harbin 150040, China

**Keywords:** *Betula pendula*, tetraploid, leaf morphology, growth polarity, leaf size, cell expansion

## Abstract

Plant polyploidization changes its leaf morphology and leaf development patterns. Understanding changes in leaf morphology and development patterns is a prerequisite and key to studying leaf development in polyploid plants. In this study, we quantified and analyzed the differences in leaf morphology, leaf growth polarity, and leaf size between diploid and tetraploid birches (*Betula pendula subsp. pendula*), and preliminarily investigated genes involved in leaf growth and development in birch. The results showed significant changes in leaf morphology in tetraploid birches, especially the basal part of the leaf. In addition, the proximal growth rate of tetraploid leaves was altered. The changed proximal growth rate did not affect the growth polarity pattern of tetraploid leaves. The leaf area of tetraploid was significantly larger than that of diploid birch. The difference in leaf size was mainly due to differences in their growth rates in the middle and late stages of leaf development. Increased cell expansion capacity was the major reason for the enormous leaves of tetraploid birch; however, cell proliferation did not contribute to the larger tetraploid leaf. The gene expression of *ATHB12* was associated with cell size and leaf area, and may be a critical gene affecting the leaf size in diploid and tetraploid birches. The results will provide valuable insights into plant polyploid leaf development and a theoretical basis for later investigations into the molecular mechanisms underlying the gigantism of tetraploid birch leaves.

## 1. Introduction

Plant polyploids are new species formed when plant genomes are polyploids, often containing multiple genomes. Previous studies have shown that after plant polyploidy, each gene’s gene dose and total DNA content are doubled. These two effects directly or indirectly increase the size of cells and organelles by changing gene expression [1], resulting in many changes in polyploid plants’ growth and physiological characteristics. Among them, as important organs for sensing the external environment and coordinate plant growth, the morphological changes of leaves are particularly significant. The leaves of tetraploid *Arabidopsis thaliana* have larger leaves [2,3]. The leaf shape index (leaf length/width) of the tetraploid drumstick tree (*Moringa oleifera Lam*.) was lower than that of diploid plants [4]. The leaf length, width, and thickness of autopolyploid *Chrysanthemum* lavandulifolium (*Fisch.ex Trautv*.) increased [5]. Poplar triploid [(*P. simonii* × *P. nigra var. Italica*) × (*P.* × ‘*popularis*’)] has a larger leaf size [6], but the leaf size of poplar octoploid (*P. hopeiensis*) is smaller [7].

Studies on the growth development of tetraploid birch have revealed differences in leaf morphology between tetraploid and diploid birches [8]. Recent studies have found it difficult to determine the ploidy level of birches with traditional morphological methods [9]. However, the method of using modern geometric morphological measurement methods of landmarks to carry out geometric morphological analysis of leaves has been widely used. This method can more accurately describe leaf shape differences, and effectively classify species or genotypes according to leaf shape differences [10]. North American Grapevines (*Vitis vinifera L*.) [11], *Populuswulianensis and P. ningshanica* (*Salicaceae*) [12], Chinese oaks (*Quercus L.*, *Fagaceae*) [13], and *Asteropyrum Peltatum* (*Franch.*) *Drumm. et Hutch* [14] have all been assessed in this way for variation in leaf morphology across species/genotypes.

Leaf polarity development, an important aspect of leaf morphogenesis [15,16], refers to the leaf’s proximal or distal allometric growth. The allometric growth is mainly related to the changes in spatial and temporal patterns of cell division and differentiation [17]. Depending on the proximal and distal growth rates of the leaf, the growth pattern of the leaf is classified as positive allometry (basipetal growth), negative allometry (acropetal growth), isometry (diffused/even growth), and complex allometry or bidirectional growth [17,18]. Previous studies have shown that closely related species may also show changes in growth polarity patterns [18].

Leaf growth and expansion is another important aspect of leaf morphogenesis. It refers to the process by which the leaf primordium reaches its final size under cell proliferation and expansion control [16,19]. At the cytological and molecular levels, the final size of plant leaves depends on the size and number of cells produced [20]. The relationship between cell size, cell number, and leaf area has been demonstrated in many plants. Leaflet mutants screened in *Arabidopsis thaliana* generally had fewer cell numbers and smaller cell sizes. Large leaf mutants are accompanied by increased cell numbers and cell sizes [21]. Several genes have been shown to increase leaf size by promoting cell proliferation. For example, the *GROWTH-REGULATING FACTOR 5*
*(GRF5**)* can increase cytokinin content and promote cell proliferation by regulating the expression of the cytokinin-related gene *CYTOKININ OXIDASE/DEHYDROGENASE1* (*CKX1**)*, thereby increasing leaf size [22,23]. *SWITCH/SUCROSE NONFERMENTING 3C* (*SWI3C**)*, *ANGUSTIFOLIA 3* (*AN3**)* and *BR INSENSITIVE 1* (*BRI1**)* have also been shown to promote leaf growth by promoting cell proliferation in plants [24,25,26,27,28]. Compared with cell division, cell expansion affecting leaf growth seems to be complex. Some key genes that affect leaf size by regulating cell expansion have been identified. For example, *ARABIDOPSIS THALIANA HOMEOBOX 12*
*(ATHB12**)* overexpression induced the early arrest of cell division and promoted the transition to endoreduplication, and cell expansion was promoted by controlling the expression of cell wall-associated genes and brassinolide biosynthetic genes, causing the plant to form larger cells and leaves [29,30]. *GRF5* has also been proven to promote poplar leaf growth by promoting cell expansion [23]. In addition, there may be a compensation system in plant leaf cells, which means that the reduction in cell number may lead to cell expansion [31].

In this study, we quantified leaf differences between diploid and tetraploid birches in terms of three aspects: leaf morphology, polarity of leaf growth, and leaf size. We focused on the factors affecting leaf size and measured the related gene expression levels affecting leaf growth and development. The study will lay a theoretical foundation for the later study on the molecular mechanisms of diploid and tetraploid birch leaves development.

## 2. Results

### 2.1. Differences in Leaf Morphology between Diploid and Tetraploid Birch

The result from Ploidy Analyzer found that the two standard peaks of three diploid birches appeared at about channels 45 and 90 with relative fluorescence intensity. The two standard peaks of three tetraploid birches appeared at about channel 90 and 180, which indicated that their DNA content was twice that of diploid birch (Figure 1A). We analyzed whether the switch from cell division to the endocycle program was altered in the tetraploid plants. The results showed that the peak plots of the tetraploids showed no significant changes in their ploidy ratios relative to diploids (Appendix A). This indicated that the tetraploid birch did not alter endoreduplication onset. The ploidy level of birch used in this study was accurate and can be used for subsequent experiments. The morphology of the leaves changed after doubling the birch chromosomes (Figure 1B). In order to identify the morphological differences in the leaves, 13 landmarks were marked on the birch leaves in this study (Appendix A) and were converted to a configuration of 26 cartesian coordinates in 13 pairs (x, y) for each leaf. The raw coordinate matrix obtained was converted to a normalization matrix by Procrustes fitting. The configuration was rotated to a central distribution around the 13 leaf landmarks based on the raw coordinate matrix (Figure 1C).

The results of the PCA showed that the first five principal components accounted for 93.18% of all leaf shape variations (Appendix A). Of these, PC1 (70.35%) showed the main differences in leaf shape, and distinguished diploids and tetraploids; PC2 (9.63%) showed the secondary differences in leaf shape, which revealed the variation among different leaves of birch (Figure 2A). Visualization of the results from PC1 showed that the leaf shapes of diploid and tetraploid birches largely matched. Morphological variation between leaves of different ploidy mainly occurred at the leaf base. The leaf base of diploid birch was truncate, while that of tetraploid birch was cordate (left side of Figure 2B). The widest position of the diploid birch leaf blade was above the lower 1/5th position. In comparison, the widest position of the tetraploid birch leaf blade was below the lower 1/5th position (left side of Figure 2B). Visualization of PC2 revealed morphological variation in the apical part of birch leaves. The variation was mainly shown as whether the leaf apex was wider or narrower, but it was not related to the ploidy level (posterior side of Figure 2B).

Overall, according to the PCA results, the morphological variation of diploid and tetraploid birch leaves accounted for 70.35% of the total morphological changes. The significant difference between diploid and tetraploid leaves of birch was mainly reflected in the leaf base.

### 2.2. No Change in the Polarity Pattern of Leaf Growth in Tetraploid Birch

Leaves of different plants often show characteristic growth polarity patterns as they develop. In order to investigate the growth polarity patterns of diploid and tetraploid leaves of birch, we marked ink dots at the center of young leaves (Figure 3A,B) and substituted the regularly measured (xi, yi) into the formula to obtain the growth ratio (α) (Figure 3C,D). The results showed that α was greater than 1 for both diploid and tetraploid birch leaves. So the growth pattern of birch leaves was a positive allometric pattern of leaf base growth. The α of diploid birch leaves was 1.1477 ± 0.0996, and that of tetraploid birch leaves was 1.4344 ± 0.17504 (Figure 3C,D), indicating that the growth ratio (α) of tetraploid birch leaves was slightly more significant than that of diploid birch.

Further research showed that the absolute growth rate (AGR) of proximal (y) of tetraploid leaves was significantly higher than that of diploid leaves (*p* < 0.01). However, there was no significant difference in the absolute growth rate (AGR) of distal (x) (Appendix A). Therefore, the higher growth ratio (α) of tetraploid leaves was due to the faster growth rate of proximal (y). Although the growth ratio (α) of diploid and tetraploid leaves was slightly different, the analysis of variance showed that in the growth ratio (α) of the two, there was no statistical significance (Figure 3E). Thus, there was no significant change in the growth ratio of tetraploid leaves of birch.

Compared with diploid, the proximal (y) growth rate of tetraploid birch leaves was higher, resulting in a larger growth ratio (α). However, the growth ratio (α) of tetraploid birch was not significantly different from that of the diploid leaves, and there was no change in leaf growth polarity patterns of tetraploid birch.

### 2.3. The Leaf Area of Tetraploid Birch Is Significantly Larger Than That of Diploid

It has been previously reported that tetraploid birch has a larger leaf area than diploid [8]. Similarly, we found that the leaf area of three tetraploid birches was significantly larger than that of three diploid birches by ANOVA analysis on the leaf area of six trees (Figure 4A). In addition, the average leaf area of tetraploid Te-1 was 2.54 times, 1.99 times, and 1.69 times of that of three diploid trees, respectively. Moreover, compared with diploid birches, tetraploid birches had longer, wider leaves and an increased leaf index (Figure 4B). In addition to this, the differences in leaf length and width between tetraploid Te-3 and diploid Di-3 were not significant (Figure 4B). However, the difference in leaf area between these two was significant (Figure 4A). It was speculated that this may be due to the variation in the leaf base of tetraploid leaves, which increases the leaf area of tetraploids (Figure 2B).

It is now known that the increase in leaf area may be due to larger leaf primordia, faster growth rates, or prolonged growth periods during meristem initiation [28]. In order to find the developmental stage that affects the leaf area change in tetraploid birch, we performed leaf growth analyses on the leaves of six trees. The results showed that birch leaves developed slowly in the early stages (0–24 days), then grew rapidly and reached maximum leaf area at around 40 days (Figure 4C). Comparing the growth curves during the first 0–12 days, diploid and tetraploid leaves sizes were similar; from 16d on, the difference in leaf area among different ploidy began to appear, and the growth rate of the leaf was much higher in tetraploids than in diploids, which finally resulted in tetraploids possessing a larger leaf area.

### 2.4. Cell Expansion Causes the Large Size of Tetraploid Birch Leaves

The leaf size is determined by its cell size and number [20]. Although tetraploid birch leaves are larger than those of diploids, it is not clear whether this phenomenon is caused by the enhanced cell division ability or the enhanced cell expansion ability. Therefore, we observed and counted palisade cells and the lower epidermis cells after birch leaves were treated to transparency.

The results showed that the palisade cell area was larger in the tetraploid than in the diploid leaf. In addition, the difference in palisade cell area between the two species was very significant (Figure 5A,B). The average palisade cell area of tetraploid Te-1 was 1.87, 1.69, and 1.60 times of that of the three diploids, respectively. Furthermore, the average palisade cell area of the other two tetraploids, Te-2 and Te-3, was also more than 1.45 times that of the diploids (Figure 5A,B). Similarly, after observing the lower epidermis cells of the two species, it was found that the area of the lower epidermis cells of the tetraploid was significantly larger than that of the diploid (Figure 5C,D). The average epidermal cell area of tetraploid Te-2 was 2.07 times, 1.76 times, and 1.46 times of that of three diploids, respectively. The average epidermal cell area of Te-1 and Te-3 was more than 1.33 times that of the diploids. Therefore, it can be concluded that leaf cells of tetraploid birch had stronger cell expansion capacities than those of the diploid birch. Furthermore, there was a significant positive correlation between palisade cell area and leaf area, as well as between epidermal cell area and leaf area (Figure 5E,F), i.e., a leaf with a larger cell area had a larger leaf area. Therefore, increased ploidy level enhanced the birch’s cell expansion ability and enlarged its cell area, meaning that it is the main reason for the larger size of tetraploid birch leaves.

Based on the average area of palisade cells and epidermal cells, we calculated the number of leaf cells (Figure 6A,B). The results showed that there was no significant difference in the number of leaf cells between diploid and tetraploid birches, and that even tetraploid Te-2 and Te-3 had significantly fewer leaf cells than diploid Di-2 and Di-3. Therefore, the larger leaves of tetraploid birch were not caused by the change in cell number.

To this end, we further assayed the relative gene expression of the cell cycle regulatory genes *PROLIFERATING CELLULAR NUCLEAR ANTIGEN 1*
*(PCNA1**)* (Figure 6C), *CYCLIN D3*
*(CycD3**;1)* (Figure 6D), *CYCLIN-DEPENDENT KINASE B1;1*
*(CDKB1**;1)* (Figure 6E) and *CYCLIN-DEPENDENT KINASE B2;1*
*(CDKB2**;1)* (Figure 6F) in the apical buds and leaves of diploid and tetraploid birches. These genes are known to be closely associated with cell division [32,33,34,35], and can represent cell proliferation ability to some extent [36,37]. The results showed that these four genes were more highly expressed in the apical buds of Di-2, Di-3, and Te-1 with more leaf cells, indicating that these three trees had a stronger cell proliferation capacity. However, there was no difference in the expression level of these four genes among different ploidy of birch. The results of transcriptome data analysis also showed that although the expression of four genes in diploid and tetraploid birch was slightly different, their changing trends did not show consistency (Appendix A). Therefore, the cell proliferation capacity did not differ between ploidy levels and did not affect the final leaf area. Thus, it can be concluded that no significant changes in cell numbers occurred after chromosome doubling in birch. Cell proliferation was not the major cause of the larger leaves of tetraploid birch.

### 2.5. Gene Expression of the ATHB12 Is Significantly Correlated with Leaf Size in Birch

We analyzed our previous transcriptome data on diploids and tetraploids. In the biological regulation and cellular process enrichment pathways in 4x vs. 2x up-regulated DEGs enriched GO analysis (Appendix A), we found three genes that have been shown to be involved in regulating leaf development. These three genes all had higher expression in tetraploids (Appendix A). Among them, *SWI3C* (Figure 7A) and *GRF5* (Figure 7B) were shown to increase leaf size by promoting cell proliferation. *GRF5* (Figure 7B) and *ATHB12* (Figure 7C) were shown to increase leaf size by promoting cell expansion. Therefore, we determined the relative expression level of these four genes in the apical buds and first leaves of three diploid birches and three tetraploid birches.

Our results revealed that both genes related to cell proliferation were all highly relative, expressed in the apical buds of Di-1, Di-2 and Te-1 with more leaf cells, indicating that they may promote the leaf cell proliferation of these three trees to some extent, but this did not affect the final size of their leaves. However, the expression level of the *ATHB12* was significantly correlated with the size of birch leaves. Firstly, the relative expression of the *ATHB12* was significantly higher in both the apical bud and first leaf of the tetraploids than in the diploids, suggesting that the relative expression of this gene may be directly related to birch ploidy. Secondly, correlation analysis showed that the relative expression of the *ATHB12* was significantly and positively correlated with both cell area and leaf area (Figure 7D,E). In addition, there was a consistent trend of *ATHB12* relative expression with cell size and leaf area: trees with lower relative expression of *ATHB12* had smaller cell areas and smaller leaves; conversely, trees with higher relative expression of *ATHB12* had larger cell areas and larger leaves (Figure 7F). Therefore, we speculate that doubling of birch chromosomes may promote the expression of *ATHB12*, and further promote the expansion of leaf cells, which ultimately results in larger cell areas and leaf areas of tetraploid birch.

## 3. Discussion

The leaves of polyploid plants often have typical morphological characteristics, which can be used as an important basis for rapid and efficient identification and differentiation of plant ploidy [2,38]. Recent studies have shown that the ploidy level of birches are not effectively distinguished by measuring and counting traits such as leaf length, width, and tip angle [9]. However, *Betula albosinensis* (red birch) and *Betula platyphylla* (white birch) can be classified into species based on leaf morphology [39]. Thus, in this study, principal component analysis (PCA) was performed on diploid and tetraploid birches by converting the position of the leaves into landmarks and visualizing the morphological variation. We found that there were no intersections between leaf collections of diploid and tetraploid birches. Morphological differences between the two species were distinguished by PC1 (70.35%), and such differences in morphology were mainly at the leaf base. Specifically, the leaf bases changed from truncate to cordate after polyploidization of the birch, with the widest position of the leaf moving towards the petiole. This variation at the leaf base is an important reason for the larger leaf area of the tetraploid birch. Therefore, there were significant differences in leaf morphology between diploid and tetraploid birch. The ploidy of birch can be preliminarily screened and evaluated through this morphological difference, which will significantly simplify the identification steps after polyploid induction of birch.

It has been shown that the cell proliferation process gradually stops during leaf development from the distal to the proximal end of the leaf [19]. The proximal part of the young leaf is the main site of cell proliferation. Our study revealed that the proximal part of the leaf shows significant morphological variation in tetraploid birches compared to diploids (Figure 2B). Thus, we suspect that the cell proliferation pattern may be altered in the tetraploid birch, and may lead to a change in the polarity pattern of its leaf growth. For this reason, we investigated the growth polarity patterns of diploid and tetraploid birches. The results showed that the growth polarity pattern of both diploid and tetraploid was a positive anisotropic basal growth (α > 1) and that the growth ratios did not show significant differences. Thus, birch polyploidization did not show a change in growth polarity pattern. However, our calculation of the absolute growth rate at the proximal and distal ends of leaves revealed that the proximal growth rate was faster in tetraploid birch leaves. We speculate that this may be an important reason for the higher leaf growth rate and the variation of leaf base morphology of tetraploid birch.

Previous findings showed that tetraploid birch possesses a large leaf area [8]. In this study, we found that the leaf area of tetraploid birch was significantly larger than that of diploid birch. However, the leaf length and width of diploid and tetraploid leaves basically grew synchronously. We believe that the slightly different leaf index is due to individual differences, not ploidy changes. Thus, tetraploid birches did not show a change in leaf length/leaf width ratios. This was consistent with the results of the PCA visual model. Leaf growth analysis revealed that diploid and tetraploid birch leaves grow in an ‘S’-type model, and the growth rate in the middle to late stages of development determines their final leaf size differences. Therefore, tetraploid birch leaves possessed a higher growth rate than diploid, which was an important reason for the large leaf size of tetraploid birch.

In order to find the key factors affecting the size difference of diploid and tetraploid birch leaves at cytological and molecular levels, we measured and counted the area and number of palisade and epidermal cells of birch leaves. The results showed that the area of epidermal cells and palisade cells in tetraploid birch was significantly larger than in diploid birch. Although previous studies have shown a strong correlation between ploidy and cell volume in pavement cells of the epidermis, this correlation was extremely weak in palisade mesophyll cells [40,41]. However, our study has a different finding, that ploidy level promotes not only the size of epidermal cells, but also that of palisade cells. However, there was no significant difference in the total number of cells between diploid and tetraploid leaves, and even the leaf cell numbers of the two tetraploids showed a decrease in cell numbers. Previous studies similarly showed that an increase in ploidy would result in a larger cell area and lower cell numbers of *Arabidopsis* [42]. It was also further verified that the cell proliferation ability showed no difference between ploidy levels by measuring the amount of cell division-related gene relative expression. Therefore, cell expansion is an important reason for the larger leaves of tetraploid birch.

It is known that there is a significant correlation between ploidy level and cell size [40], but the reason for the increased cell size according to ploidy level is currently undefined. One explanation is that an increase in ploidy directly affects the cell size of plants, increasing cell area by approximately 1.58-fold [43]. The change is mainly due to the increased content of DNA that causes the nucleus to become larger [44,45], but is independent of gene regulation. However, it has also been suggested that genes such as *EXPANSIN 10*
*(EXP10**)* [46], *GROWTH-REGULATING FACTOR 2*
*(GRF2**)* [47], and *ARGOS-LIKE* (ARL) [48] can also contribute to leaf growth by promoting cell expansion of plants, so the increase in ploidy level may not be directly related to cell volume, and ploidy level may indirectly control cell expansion or the magnitude of cell expansion through gene regulation [49]. Recently, the inference of the latter has been demonstrated: the increased ploidy level can promote cell expansion and cell proliferation of triploid poplar by regulating GRF5 expression, thus resulting in the triploid poplar having larger leaves [23]. Therefore, to find the essential genes affecting cell size and leaf size in tetraploid birch, the expression levels of three genes known to regulate leaf development were measured in this study. The results showed that *GRF5* and *SWI3C* expression was not significantly different between diploid and tetraploid apical buds, but was highly expressed in the apical buds of trees with higher cell numbers. Thus, these two genes may have contributed, to some extent, to cell division in the leaves of these trees. An increased number of cells is not the main reason for the bigger leaves. Our results showed that the expression of the *ATHB12* was significantly higher in the apical buds and young leaves of tetraploids than in diploids. The expression level of this gene was significantly correlated with cell area and leaf area. Studies in *Arabidopsis* also showed that the *ATHB12* could promote leaf growth by promoting cell expansion as a downstream regulator of *CIN-like TCP13* [30]. Therefore, we speculate that the *ATHB12* is influenced by the ploidy level and promotes cell expansion of tetraploid birch leaves, ultimately resulting in a larger cell and leaf area of tetraploid birch. Previous studies have also shown that the overexpression of *ATHB12* could make the plants have larger leaves. However, it also could reduce the content of GA9 by inhibiting the expression of *GA20*
*OXIDASE1* (*GA20ox1**)*, thus inhibiting the growth of stems and leading to significant dwarfing of plants [50]. It is consistent with the phenotype of tetraploid birch [8]. Similarly, we found that the *GA20ox1* gene was expressed at lower levels in the apical buds of tetraploid birch by analysis of transcriptome data (Appendix A). Because the expression of *ATHB12* in the apical buds of tetraploids was significantly higher than that in diploids, we speculated that the higher level of expression of the *ATHB12* affected the accumulation of gibberellin content in the apical buds of tetraploid birch after the doubling of birch chromosomes, which resulted in the slower growth rate of the shoot tips of tetraploid birch.

## 4. Materials and Methods

### 4.1. Plant Material

The experimental material consisted of three tetraploid and three diploid birches that have been planted in the birch breeding base of Northeast Forestry University (126°64′ E, 45°72′ N) for 18 years. The three selected diploid birches were excellent cultivars and breeding parents. The tetraploid birches were obtained by soaking birch seeds with colchicine.

### 4.2. Determination of Ploidy Levels in Birch

Young leaves were collected, and midrib and leaf margin serrations were removed. Approximate 0.5 cm × 0.5 cm leaf pieces were placed in a glass culture dish with 2 mL of pre-cooled extraction buffer [51]. The leaf flesh was quickly chopped vertically with a single-peaked blade, and 1 mL of DAPI staining solution (Sysmex, Kobe, Japan) was added. Leaf flesh was filtered through a 30 μm cell sieve into a sampling tube, and ploidy was measured using Ploidy Analyzer (PA, Partec, Germany) [8].

### 4.3. Measurement of Leaf Dimensional Blade Parameters

At the beginning of June 2022, 26 fifth mature leaves were picked from the branches at the same position of each of the three tetraploid and three diploid birches. Leaves were brought back to the laboratory and imaged with a scanner. The leaf length, width, and area were measured by ImageJ software [52].

### 4.4. Leaf Morphometric Analysis

Twelve leaves from each tree were collected and measured. Thirteen landmarks were selected from each scanned leaf [39]. The leaf base was the origin, and the line connecting the leaf tip and the leaf base was the y-axis. ImageJ measured each position’s x and y coordinates. The 13 pairs of (x, y) coordinates were obtained and subjected to Procrustes analysis (GPA) using the MorphoJ software (https://morphometrics.uk/MorphoJ_page.html, accessed on 16 June 2022) [53]. Principal component analysis (PCA) was performed using the data from the normalized matrix [53].

### 4.5. Leaf Growth Analysis

In the middle of May 2022, the position of 9 first leaves of each tree was marked. The leaves were photographed every 4 days for 52 days, and the leaf area was measured by the ImageJ until the marked leaves were utterly mature. The “S” curve was drawn according to leaf area and time.

### 4.6. Determination of the Size and Number of Leaf Cells

The mature leaves of the sampled trees were fixed in FAA solution for 24–48 h and then incubated in chloral hydrate solution (chloral hydrate 200 g; glycerin 20 g; water 50 mL) for transparency. Afterward, the subepidermal palisade cells between the leaf midrib and margin were observed by an optical microscope (OLYMPUS BX43F). The area of palisade cells was measured by the ImageJ program. Six leaves were selected for each tree, and each leaf’s area of 40 cells was measured. The leaf area was divided by the cell area in order to calculate the number of cells in palisade cells under the epidermis.

Leaf tissues were fixed in fixative solution (ethanol/acetic acid = 9/1) for 24 h and soaked in chloral hydrate solution for 10 min. The leaf’s lower epidermal cells were observed by an optical microscope (OLYMPUS BX43F). The lower epidermal cell area was measured by the ImageI program. Six leaves were selected for each tree, and each leaf’s area of 24 cells was measured. The cell number was calculated as descripted above.

### 4.7. Measurement of Leaf Growth Ratio (α)

At the start of the measurements in mid-May 2022, 8 young leaves of each tree were marked with an ink dot at the middle of the basal apical axis. The length of this mark to the apical part of the leaf (x) and the basal part of the leaf (y) was equal, and nine leaves were marked for each sampled tree. The x and y were measured every eight days during leaf growth, and the four sets of (xi, yi) values obtained were fitted to the power equation y = bxa and plotted on a double logarithmic grid to calculate the growth ratio (α) [18].

The growth polarity patterns of plant leaves can be divided into three types according to the growth rates of proximal (y) and distal (x) leaves. If x equals y (α = 1), leaves grow in equal length; if x is greater than y (α < 1), the leaves grow at a negative allometric rate; if x is less than y (α > 1), then the leaves grow at a positive allometric rate [18].

### 4.8. qRT-PCR Analysis

The birch apical buds and first leaves were picked in early June 2022, and were wrapped in tin foil and frozen rapidly in liquid nitrogen. Total RNA was extracted using an RNA extraction kit (BioTeke Corporation, Beijing, China) and reverse transcribed into cDNA (ReverTreAce ^®^ qPCR RT Kit, Toyobo, Osaka, Japan). qRT-PCR (SYBR Green PCR master mix, Toyobo, Osaka, Japan) was completed using the ABI 7500 Real-Time PCR System. The ACTIN and 18s genes were used as internal controls to normalize the total RNA level present in each reaction. Expression levels were determined according to the 2^−ΔΔCt^ method. Three independent biological replicates were used in the analysis.

All gene sequences were obtained by aligning the Arabidopsis thaliana gene sequence (https://www.arabidopsis.org/, accessed on 4 May 2022) with the birch reference genome (https://genomevolution.org/CoGe/GenomeInfo.pl?gid=35080, accessed on 4 May 2022). Appendix A shows the gene names and primer sequences.

### 4.9. Pearson Correlation Analysis

Pearson correlation analysis was performed using SPSS 26.0 software.

### 4.10. RNA-Seq Experiment and Data Analysis

We analyzed previous RNA-seq data of the apical buds of 8-year-old diploid and tetraploid birches. Plant samples and sampling methods have been described previously [8]. The clean data were aligned to the birch reference genome (https://genomevolution.org/CoGe/GenomeInfo.pl?gid=35080, accessed on 26 June 2022) using Hisat2- 2.1.0 (https://github.com/DaehwanKimLab/hisat2, accessed on 26 June 2022) [54]. The count value for each gene was calculated using Stringtie- v2.2.0 (http://ccb.jhu.edu/software/stringtie, accessed on 26 June 2022) [55]. Differentially expressed genes (DEGs) were identified with threshold of |log2FoldChange| ≥ 1. DEGs were used for gene ontology (GO) enrichment analysis using the GO enrichment tool (http://geneontology.org, accessed on 30 July 2022) with *p*-value less than 0.01 as the threshold for significant enrichment (Appendix A).

## Figures and Tables

**Figure 1 ijms-23-12966-f001:**
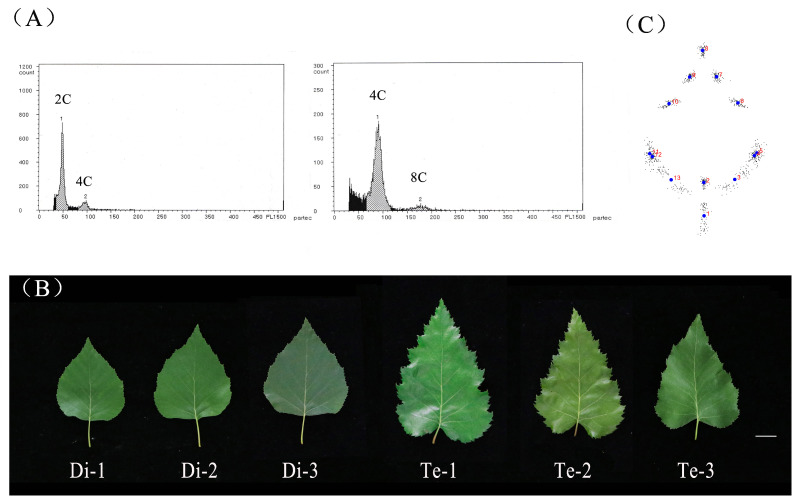
Phenotypic analysis of diploid and tetraploid birches. (**A**) Ploidy analyzer profiles of DAPI-stained leaf nuclei of diploid birch (left) and tetraploid birch (right). The x-axis indicates the fluorescence intensity of DAPI, which reflects the nuclear DNA content. (**B**) The 5th mature leaf of diploid and tetraploid birches, bar = 2 cm. Di, diploid. Te, tetraploid. (**C**) Leaf shape model by procrustes fit. Dots in blue represent the locations of 13 landmarks after matrix normalization, and numbers in red represent 1–13 landmarks.

**Figure 2 ijms-23-12966-f002:**
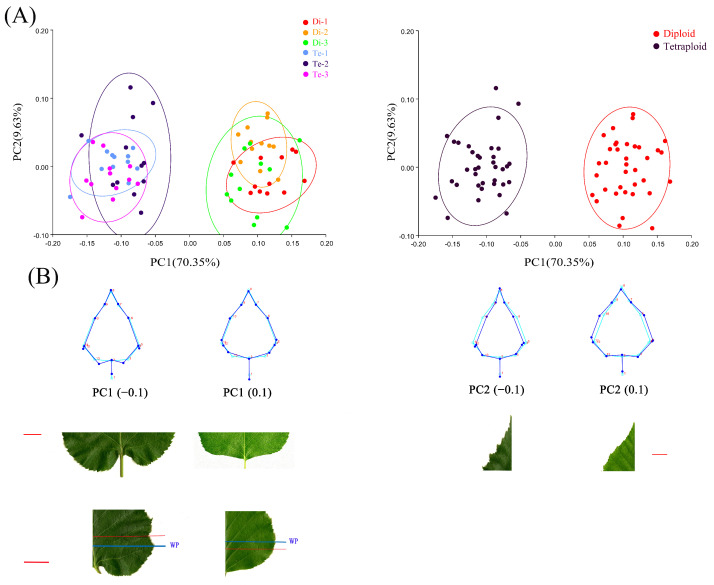
Principal component analysis (PCA) of diploid and tetraploid birch leaves. (**A**) Scatter plots of PC1 and PC2 scores for symmetric components. (**B**) Wireframe plots of different scores for PC1 and PC2 and photographs of leaves, bar = 2 cm. WP indicates widest position of leaves.

**Figure 3 ijms-23-12966-f003:**
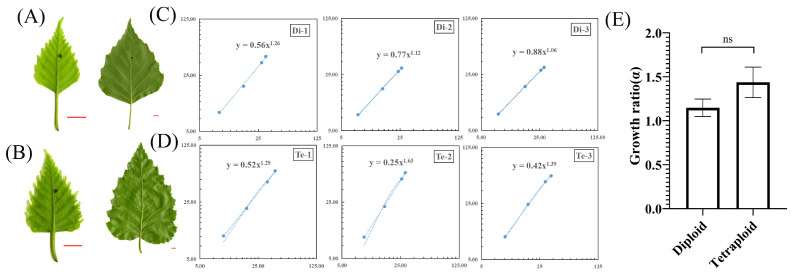
Growth polarity of diploid and tetraploid birch leaves. (**A**) Young diploid birch leaf (left) and mature leaf (right), bar = 0.5 cm. (**B**) Young tetraploid birch leaf (left) and mature leaf (right), bar = 0.5 cm. (**C**) Double log network plot of (xi, yi) obtained from measurements of diploid birch leaves, *n* = 8. (**D**) Double log network plot of (xi, yi) obtained from measurements of tetraploid birch leaves, *n* = 8. (**E**) Comparison of growth ratios (α) of diploid and tetraploid leaves, *n* = 3, ns = not significant (paired Student’s *t*-test).

**Figure 4 ijms-23-12966-f004:**
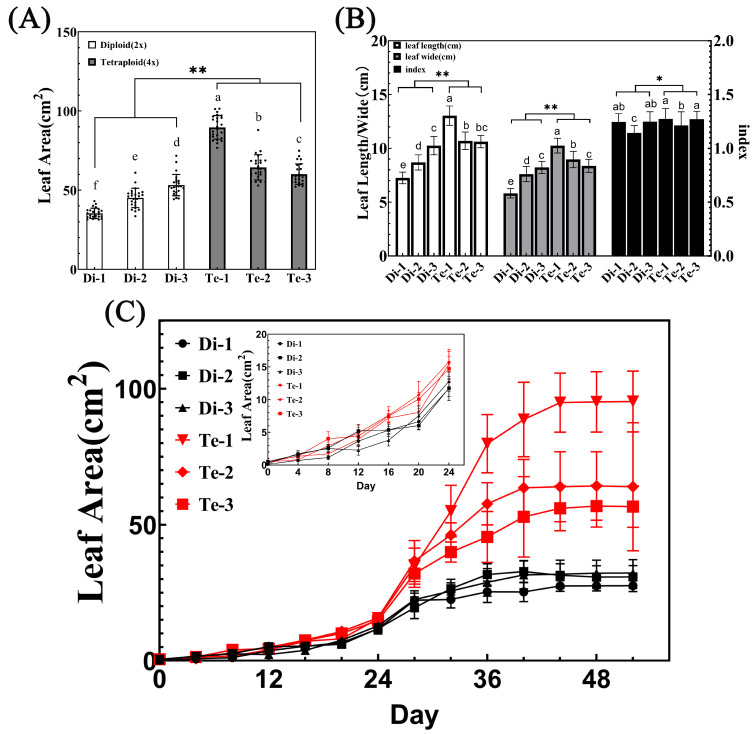
Leaf area analysis of diploid and tetraploid birches. (**A**) Leaf area of diploid and tetraploid birches, *n* = 26. Different letters denote statistically significant differences after one-way ANOVA. ** = *p* < 0.01 (paired Student’s *t*-test). (**B**) Leaf width, length, and index of diploid and tetraploid birches, *n* = 26. Different small letters denote statistically significant differences after one-way ANOVA. ** = *p* < 0.01, * = *p* < 0.05 (paired Student’s *t*-test). (**C**) Leaf growth analysis of diploid and tetraploid birches (*n* = 9).

**Figure 5 ijms-23-12966-f005:**
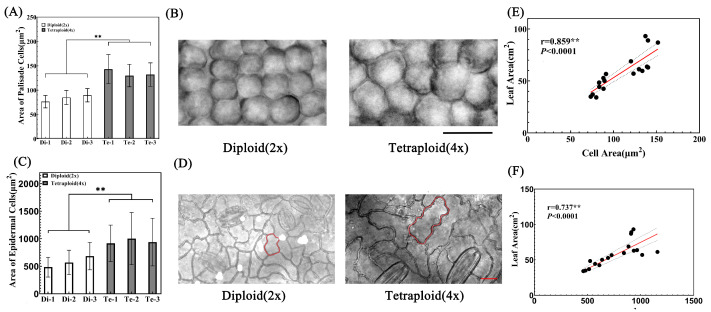
Analysis of leaf palisade cells and lower epidermal cells. (**A**) Analysis of palisade cell area of diploid and tetraploid birch leaves, *n* = 240. ** = *p* < 0.01 (paired Student’s *t*-test). (**B**) Subcutaneous view of diploid and tetraploid birch leaf palisade cells, bar = 20 μm. (**C**) Lower epidermis cell area of diploid and tetraploid birch leaves, *n* = 144. ** = *p* < 0.01 (paired Student’s *t*-test). (**D**) View of lower epidermis cells of diploid and tetraploid birches, bar = 20 μm. (**E**) Correlation analysis between palisade cell area and leaf area. (**F**) Correlation analysis between lower epidermis cell area and leaf area.

**Figure 6 ijms-23-12966-f006:**
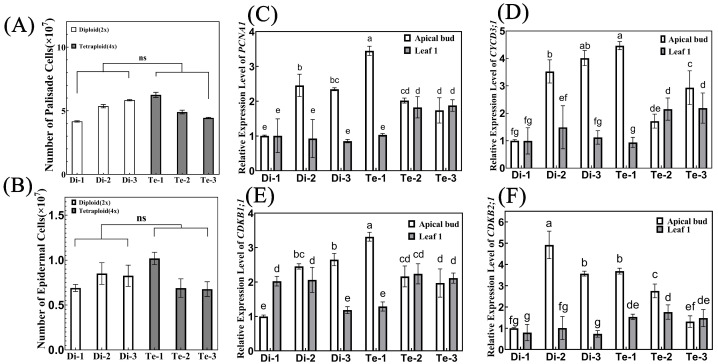
Number of leaf cells and relative gene expression levels of cell cycle-related genes. (**A**,**B**) The number of palisade cells (**A**) and lower epidermal cells (**B**) in diploid and tetraploid birch leaves, n = 6. ns = no significance (paired Student’s *t*-test). (**C**–**F**) Relative expression of *PCNA1* (**C**), *C**YCD3**;1* (**D**), *CDKB1**;1* (**E**), and *CDKB2**;1* (**F**) in the apical bud and first leaf of diploid and tetraploid birches, *n* = 3. Different letters denote statistically significant differences after one-way ANOVA.

**Figure 7 ijms-23-12966-f007:**
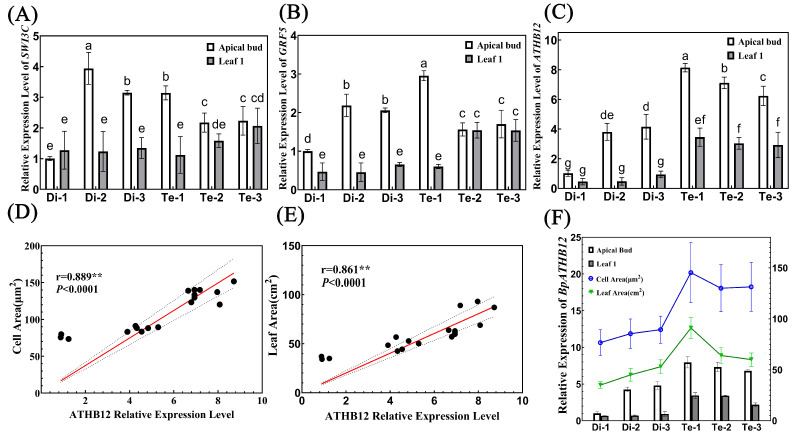
Analysis of the relationship between the expression of genes and leaf areas. (**A**–**C**) Relative expression of *SWI3C* (**A**), *GRF5* (**B**), and *ATHB12* (**C**) in diploid and tetraploid birches, *n* = 3. Different letters denote statistically significant differences after one-way ANOVA. (**D**) Correlation analysis between *ATHB12* relative expression and cell area. (**E**) Correlation analysis between *ATHB12* relative expression and leaf area. (**F**) A combined graph of the *ATHB12* relative expression, cell area, and leaf area.

## Data Availability

The original contributions presented in the study are included in the article/Appendix A, further inquiries can be directed to the corresponding authors.

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
