# Peer review of "Differences in Leaf Morphology and Related Gene Expression between Diploid and Tetraploid Birch (Betula pendula)"

_ijms, 2022, doi:10.3390/ijms232112966_

Round 1
Reviewer 1 Report
Zhang and co-workers have studied leaf morphology and related gene expression of diploid and tetraploid birches (Betula pendula). The manuscript is well written and carefully drafted. However, here are some comments that authors should address before the publication of the article. The methods have been described very shortly. It is not clear how the differential gene expression analyses of RNA-seq data were performed or how the specific genes were chosen for the qPCR analyses. Please provide more details for RNA-seq data analysis. Was there only one reference gene in qPCR analyses? This is not adequate. MIQE guidelines suggest the use of at least two reference genes.
Reviewer 2 Report
In this study, Zhang et al. conducted morphology analysis and gene expression for leaf development of diploid and tetraploid birch to reveal the correlation between leaf shape and polyploidization. Authors wrote the manuscript very nicely. There are still some problems before acceptance.
Specific problems:
Tense. The Result section should be described in simple past tense.
Statistics. Please use always P as italic and upper case in the text. Generally, I don't think it is appropriate to associate different P levels to different parameters, e.g., P≤0.05, 0.01, 0.001 this make the comparison of the effect of the factors very hard, so please use one level of significance. In addition, please check the significance listed in Line 168 and Fig. 4B, significant or ns.
Abbreviation. Use Latin names and abbreviations of genes correctly. Species and genes are given their full names when they first appear, followed by acronyms.
Figure legends. Please add the meanings of different color in Fig. 1C, different small letters in Fig. 6 and 7.
L18: What does this mean?
L182: It’s hard to understand 0-24 d here, please add the detailed explanation in text.
L195-196, 225: The order and the citation of each Figure must be in sequence and correct.
L221-222: Is there a one-to-one relationship between Di-1 and Te-1, Di-2 and Te-2, Di-3 and Te-3?
L246: Please describe the samples for the transcriptome data.
L348: The authors measured the expression levels of serval related gene, however, just discussed the regulation of ATHB12 during leaf development. Moreover, the expression patterns of related genes at different stages like Fig. 4C may obtain more persuasive conclusion.
L39: Is there some difference of leaf thickness found in this study?
L380, 422: Which year.
L381, 385: Please add the description of experimental replication in this study. I am very puzzled to understand the meaning of n=8, n=3, n=240, n=145, n=6.
L398-400: When writing about measurements, use a space between a number and its unit.
L438-439: The version of the used software.
Round 2
Reviewer 2 Report
In the revised manuscript, all questions (of Reviewer #2) have been addressed. Generally, I would suggest a publication of this manuscript in International Journal of Molecular Sciences.
P≤0.05 is enough for us to describing the highly or extremely significance. Anyway, all can be. One minor question is that the authors use all significantly higher to describe the results of Fig. 4B in lines 173-174. Please consider it.
Author Response
Authors Notes
We response to Reviewer’s comments in red font color.
REVIEWER 2
One minor question is that the authors use all significantly higher to describe the results of Fig. 4B in lines 173-174. Please consider it.
We thank Reviewer 2 for the comment. We have corrected it. See L173-L175.